# The Impact of Profile Genesis and Land Use of Histosol on Its Organic Substance Stability and Humic Acid Quality at the Molecular Level

Kristina Amaleviciute-Volunge [1,*], Jonas Volungevicius [2], Justinas Ceponkus [3], Rasa Platakyte [3], Ieva Mockeviciene [4], Alvyra Slepetiene [1] and Viia Lepane [5]

1   Chemical Research Laboratory, Institute of Agriculture, Lithuanian Research Centre for Agriculture and Forestry, Instituto al. 1, Akademija, 58344 Kėdainiai, Lithuania
2   Institute of Geosciences, Faculty of Chemistry and Geosciences, Vilnius University, M. K. Čiurlionio g. 21, 01513 Vilnius, Lithuania
3   Institute of Chemical Physics, Faculty of Physics, Vilnius University, Saulėtekio al. 9, 01513 Vilnius, Lithuania
4   Vezaiciai Branch, Research Centre for Agriculture and Forestry, Gargždų St. 29, 58344 Vėžaičiai, Lithuania
5   Institute of Chemistry, Tallinn University of Technology, Ehitajate tee 5, 12611 Tallinn, Estonia
*   Correspondence: kristina.amaleviciute-volunge@lammc.lt

**Abstract:** This study is designed to evaluate soil organic matter (SOM) quality indicators: molecular indicators of dissolved organic matter (DOM) and hydrophobicity of humic acid (HA), distribution of quantity in humified and labile fractions of histosols during renaturalization. The aim is to determine the differences in the qualitative composition of humic acids at the molecular level, which are decided by the previous tillage and genesis, and to evaluate the impact of anthropogenization on the peat soil according to hydrophobicity, as well as to estimate the impact of soil genesis and removing peat layer. Soil samples were taken from the three Sapric Histosol (according to WRB2022) profiles and the 0–30 cm layer in three field replicates (Lithuania, Radviliskis mun.). Our study suggested that in the differently managed drained Sapric Histosol under renaturalization, the most significant changes occurred in the topsoil layer (0–30 cm), in which an increase in the content of SOM particles 106–2 μm in size. It is expedient to grow perennial grasses and legumes to maintain the soil organic carbon stability mobile humic acids to mobile fulvic acids ratio (MHA:MFA 0.83 to 0.86). An evaluation of the quality of HA (E4:E6) revealed their highest maturity in the unfertilized perennial grasses (3.88) and crop rotation (3.87) with grasses. The highest concentrations of hydrophilic groups (ratio of the C=O to O-H) were found in Sapric Histosol under deciduous hardwood forest (12.33). The lowest hydrophilicity (9.25 and 9.36) was of the crop rotation Sapric Histosol with removed peat layer. The most sustainable use of drained Sapric Histosol in the context of the sustainability and quality of its humus substances should be associated with the formation of perennial grass and clover grassland and the cultivation of deciduous hardwood. Therefore, the horizon forms on the top part of the profile, which protects deeper Histosolic material layers from its mineralization.

**Keywords:** Sapric Histosol; E4:E6; molecular weight; hydrophobicity; FT–IR absorption

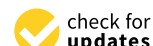



## 1. Introduction

Peat soils (Histosol) accumulate a lot of organic carbon and therefore have a significant impact on the global C cycle [1–6]. Peatlands cover only about 3% of the world's land area [4]. Natural peatlands are the largest long-term carbon storage in the terrestrial biosphere and one of the most important ecosystems contributing to climate change mitigation [7]. During photosynthesis, intact peatlands absorb 0.37 Gt of $CO_2$ from the atmosphere [2,8]. When natural peatlands are damaged, i.e., after they are drained, the opposite processes begin to take place—peatlands become a source of gases ($CO_2$, $CH_4$, $N_2O$) that cause the greenhouse effect [5,9,10]. In many peat-rich EU countries (Germany

37%, Poland 42%, Finland 62%), more than 50% of peatland is degraded—in Germany, for example, even more than 95% [11]. The degradation of peatlands ecosystems due to peatland drainage for agriculture or forestry, atmospheric pollution deposition, road construction, peat extraction, and natural resource extraction can critically degrade ecosystem services [8,12–15]. In total, 20% of peat soils in the world and 14% in Europe are used for agriculture. The great majority of peat soils used for agriculture in Europe consists of grasslands. However, there are countries where almost all organic soils are cultivated: Poland 70%, The Netherlands 85%, Germany 85%, Greece 90%, Hungary 98%, and [16]. Peat soils, their restoration, and sustainable use are, therefore, particularly important in the context of climate change. Histosols are not investigated as organic soil in the context of a change of its soil forming processes and land use, but as a peatland and its impact on environmental change. Such an opinion is formed after the studies have been reviewed.

The role of peatlands as a long-term storage of organic carbon is extremely important in the implementation of the EU's green course commitments until 2050 [17]. The EU 2030 Soil Strategy (COM (2021) 699 final) [18] is the main strategic document for the sustainable use of soils in the European region, which makes a targeted distinction between organic and mineral soils. Obligations to reduce gas emissions by 55% have been set for the member states. Lithuania is obliged to reduce greenhouse gas emissions by 9% [19,20]. The question of sustainability of Histosols' land use is emphasized in the context of climate change, but the prospects of its preservation as a pedosystem receive less attention.

Peatland restoration as a nature-based solution not only can shift degraded peatlands from a net source of $CO_2$ to a net sink within approximately 20 years [8,21–23] but can have a biophysical climate mitigation potential [24] and increase water storage [25]. Studies show that after the restoration of drained marshes, the gas regime changes in a positive direction [8,9]. Peatland restoration is evaluated in the context of $CO_2$ emissions but not in the context of the quality and functioning of the Histosols themselves. This context is characteristic of most of the analyzed studies.

In the structure of Lithuanian agricultural land, peatlands occupy only 5.66% of all land use, but the absolute majority of them (78.2%, i.e., 150,894 ha) are drained and used for arable farming [26]. These lands include not only fens but also intermediate peatlands and bogs. More than 80% of these soils are organic carbon-rich fens peat soils [26].

Intensive land drainage in the past resulted in the Baltic countries being among the ten largest European atmospheric polluters in terms of GHG emissions from damaged wetlands: Latvia ranks 5th, Estonia ranks 8th, and Lithuania ranks 9th [27]. In Lithuania, GHG emissions generated by peatlands account for about 11% of agricultural emissions. From 2015 to 2020, the emission increased by 6.7%. During the last 10 years, agricultural GHG emissions from soils grew by 40%, and the component of peatlands, due to the intensification of their plowing, occupied second place [28].

Humus substances form most of the organic components of mineral soils and play vital roles in the biogeochemical cycles [29]. The process of humification and the qualitative composition of humic acids are affected by temperature, pH, heat, water regime, etc. [30].

The humification degree (HD) (calculated indicators) and E4:E6 ratio (a spectrophotometrical measure of humification degree) are the humification indicators of soil organic matter, which reflect the changes in humus quantity, stability, and quality. HD and E4:E6 represent well secondary humification of drained peat materials and should be used in the assessment of the SOM stability and quality [31–33]. There is a lack of studies in which the qualitative changes of humus substances in the context of changes in the land use of Histosols are investigated.

Other studies show that molecular weight and its distributions are very good indicators for the characterization of persistence, absorption, hydrophobic organic partition, obsorption onto minerals, and activated carbon of humus substances [31,34]. The hydrophobicity of peat describes a condition where soil can no longer absorb water and irreversibly dries. The main factor affecting peat stability and hydrophilicity is the peat's inherent nature, which is determined by the role of functional groups –COOH, –C=O, –C-OH, and

phenol-OH, which form the active substance of organic colloid. Organic acid derivatives are the primary form of C release from the oxidation process of the carboxyl and methoxy functional groups. Functional groups in the peat, such as carboxyl and hydroxyl, are volatile and easily transform, decomposing from CHO bonds into $CO_2$ under aerobic conditions. As peat material becomes hydrophilic, $CO_2$ is released into the atmosphere [35,36]. Since organic matter determines soil water resistance which constitutes a major part of the composition of the peat soil, it is therefore important to investigate the influence of the chemical properties of peat soil on their wettability. It was established that the changes in the chemical properties of anthropogenically transformed organic material of peat soil have a significant impact on the changes in physical and hydraulic properties [16]. This transformation of peat soils remains in progress due to drainage for agriculture [37]. Similar studies of Histosols were not conducted in Lithuania.

One of the few studies conducted in Lithuania on the effects of the renaturalization of arable lands on the soil agrochemical characteristics was carried out [38]. However, these studies were carried out only in mineral soil (sandy soil) and were focused only on their general agrochemical properties. Histosols, which are used for agriculture, have been insufficiently studied [37,39,40]. On the morphology of peat soil profile and organic matter changes after drainage and during the renaturalization process, particularly little attention has been paid. The focus of our study is to investigate how different land uses affect the accumulation and transformation peculiarities of peat soil organic carbon. We also investigate the peculiarities of organic carbon value variation in the context of peat soil renaturalization, and we ask the question of how to optimize the use of peat soils to reduce GHG emissions and contribute to organic carbon conservation in them. Our study is the first attempt to evaluate SOM quality indicators, hydrophobicity, and molecular indicators of OM quantitative distribution in humified and labile fractions of peat soil during renaturalization. The aim is to determine the differences in the qualitative composition of humic acids at the molecular level, which are decided by the previous tillage and genesis, and to evaluate the impact of anthropogenization on peat soil according to hydrophobicity.

## 2. Sites and Methods

### 2.1. Research Object and the Study Site

Field experiments were conducted at the Radviliškis Experimental Station of the Lithuanian Institute of Agriculture on a fen (Sapric Histosol) with a non-removed and removed peat layer at an altitude of 120 m above the sea level (55°45′ N. 23°30′ E) (Figure 1). The treatments were investigated 12 years after the completion of the field experiment. This peat soil has formed on calcareous ground moraine loam (was washed by glacial waters), which is gley aquiclude horizon (2Cαr) and lithogenic base for peat soil formation. This low-lying fen was affected by drainage. The investigated peat soils were named, and the morphology of their profiles was identified according to the international soil classification system WRB2022 [41]. Soil samples for chemical analyses were taken from the Histosol profiles and the 0–30 cm layer in 3 field replicates according to the Table 1:

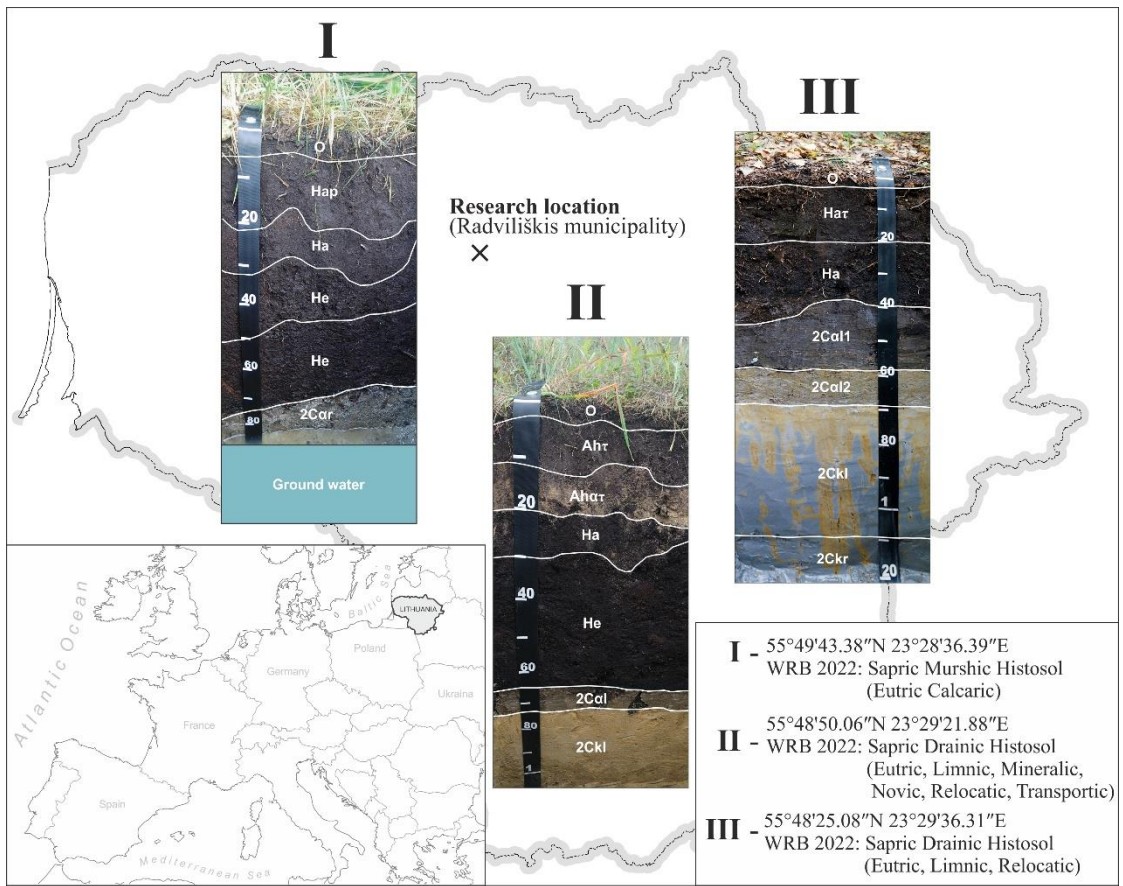

**Figure 1.** The scheme of the experimental site and sampling places: **I**—Sapric Murshic Histosol (Eutric Calcaric) with a non–removed peat layer under renaturalization; **II**—differently used Sapric Drainic Histosol (Eutric, Limnic, Mineralic, Novic, Relocatic, Transportic) with a removed peat layer; **III**—Sapric Drainic Histosol (Eutric, Limnic, Relocatic) with a removed peat layer natural forest.

**Table 1.** Investigation treatment and sampling places.

| I<br>Sapric Murshic Histosol (Eutric Calcaric) with a Non–Removed Peat Layer Under Renaturalisation (Figure 1 Part I): | II<br>Sapric Drainic Histosol (Eutric, Limnic, Relocatic) with a Removed Peat Layer (Figure 1 Part II). | III<br>Sapric Drainic Histosol (Eutric, Limnic, Relocatic) with a Removed Peat Layer (Figure 1 Part III). |
|---|---|---|
| 1. Non–used peat soil;<br>2. Previously unfertilized perennial grasses;<br>3. Previously perennial grasses fertilized with commercial NPK fertilizers *Festuca pratensis Huds+ Phleum pratense + Bromus inermis*—40% + 40% + 20%), $P_{60}K_{120}$<br>4. Previously red clover (*Trifolium pratense* L.) and timothy (*Phleum pratense* L.) mixture, $P_{60}K_{120}$;<br>5. Previously crop rotation: potatoes (*Solanum tuberosum*), winter rye (*Secale cereale*), red clover (*Trifolium pratense*), $P_{60}K_{120}$; | 1. Crop rotation field, in which there was a long–term meadow until 2001. *Fagopyrum esculentum* and *Brassica napus* were cultivated during the experimental period (2012–2014);<br>2. Sown sward, which was plowed and resown 35 years ago. The following vegetation is currently growing there: *Elytrigia repens common, Hieracium pilosella, Euphorbia cyparissias, Polygonum persicaria, Equisetum arvense, Taraxacum officinale, Achillea millefolium, Sinapis arvensis, Viola arvensis.* | 1. Natural forest, in which long-term fertilization experiments were conducted 25 years ago. Woody vegetation is prevalent: *Betula pendula, Salix caprea, Frangula alnus, Rubus idaeus, Galium mollugo* |
| The structure of the Sapric Histosol profile (according to WRB 2022) is as follows (Figure 1): | | |
| O (0–7 cm)—Hap (7–22 cm)—Ha (22–32 cm)—He (30–50 cm)—He (50–72 cm—2Cαr (>72 cm) | O (0–3 cm)—Ahτ (3–11 cm)—Ahατ (11–20 cm)—Ha (20–30 cm)—He (30–65 cm)—2Cαl (65–75 cm)—2Ckl (>75 cm). | O (0–3 cm)–Haτ (3–20 cm)–Ha (20–40 cm)–2Cαl1 (40–59 cm)—2Cαl2 (59–69 cm)—2Ckl (69–110 cm)—2Ckr (110–120 cm) |

### 2.2. Physical Methods of Analysis and Calculation

Particle size distribution of soil solid (mineral and organic) phases was determined using the light–scattering technique Mastersizer 2000 ("Malvern Instruments", Malvern, UK), which measures particles in a wide range from 2000 to 2.0 μm. Measurement is performed in liquid dispersion.

Peat soil bulk density, peat porosity, and moisture: soil samples were dry to absolutely dry masses at 105 °C 24 h and weighed to the nearest 0.001 g. According to this data, soil bulk density, moisture, and porosity were calculated by these formulas [42,43]:

Soil bulk density:

$$\rho = M_s / V_t$$

$M_S$—the weight of absolutely dry soil (dried at 105 °C), g
$V_t$—the volume of the sample (cylinder)

- Porosity: $P = [(2.63 - \rho)/2.63] \times 100$
- Moisture: $P/\rho$.

### 2.3. Methods of Chemical Analyses

Soil samples were taken from the topsoil (0–30 cm depth) (ISO 10381–4:2003) [44] in the three replicates after harvesting with a steel auger, 2.5 cm in diameter. All samples were air-dried, and visible roots and plant residues were manually removed. The samples were crushed, sieved through a 2 mm sieve, and homogeneously mixed. For the analyses of SOC, MHA, MFA, and DOC content and composition, the soil samples were passed through a 0.2 mm sieve.

Chemical analyses were carried out at the Chemical Research Laboratory at the Institute of LAMMC. Soil pH was determined in 1 M KCl (ISO 10390:2005) [45]. Total soil nitrogen (N) and the total phosphorus (P) contents were determined using a spectrophotometric measure procedure at a wavelength of 655 nm and 430 nm after mineralization with sulphuric acid ($H_2 SO_4$). The content of total potassium (K) was determined using an atomic absorption meter AAnalyst 200 (Perken Elmer, Waltham, MA, USA) [46]. Soil organic carbon content (SOC) was determined by a photometric procedure at the wavelength of 590 nm after wet combustion, according to Nikitin (1999) [47]. Mobile humus substances (MHS) and humic acids (MHA) were extracted using a 0.1 M NaOH solution and determined according to Ponomareva and Plotnikova (1980). The MFA is calculated as follows: MHS-MHA. Dissolved organic carbon (DOC) was analyzed using an ion chromatograph SKALAR (Skalar Analytical B.V., Breda, The Netherlands) ISO 8466 [48].

The humification degree (HD), according to Liaudanskiene [32] and Kalisz [33], is calculated as: HD% = HAC/SOC × 100 and was expressed as a percentage, where HAC is humic acids C content, g kg$^{-1}$; SOC—soil organic carbon content, g kg$^{-1}$. The determination of molecular indicators of dissolved organic matter (DOM) by high-performance liquid chromatography has been completed. HPLC-SEC analyses were performed at the Institute of Chemistry at Tallinn University of Technology, Estonia. The chromatograms were recorded by the Agilent ChemStation software. The polydispersity Mw/Mn was calculated from the obtained data [34]. The degree of aromaticity and condensation of humic acids is determined by the E4:E6 ratio. A total of 3 mg of HA (extracted Na OH 1N) were dissolved in 10 mL of 0.05 M Na HCO$_3$ and measuring optical density at λ = 465 nm (E4) and λ = 665 nm (E6) on UV—VIS spectrophotometer Cary 50 (*Varian*) [16,33]. Fourier transform infrared (FTIR) spectra were used to determine the HA hydrophobicity rating. FTIR spectra were recorded at the 4000–400 cm$^{-1}$ wavenumber range using an IR spectrophotometer (Verbex 70). Spectra were registered on KBr pellets obtained by pressing mixtures of 2 mg HA samples (extracted Na OH 1N) and 100 mg KBr with precautions taken to avoid moisture uptake [30,35,49].

### 2.4. Statistical Analysis

Statistical analysis was performed using the software ANOVA from the package SAS enterprise. Significant differences (one-way analysis) among treatment means were assessed by Fisher's test, and ($p < 0.05$) was considered statistically significant and is marked by letters in the text.

## 3. Results

### 3.1. Physico-Chemical Properties of Sapric Histosols Profiles and Their Morphology

The results of particle size distribution analysis showed that after domestication by draining and cultivation of Sapric Histosol with a non-removed peat layer, the content of 1000–250 µm size particles decreased, and the content of 53–2 µm size particles increased with the depth. Changes in the composition of physical particles of Histosol organic horizons highlight the line between the natural and cultivated part of the profile of Sapric Histosol. The content of particles, whose size is 250–38 µm, increased in the 0–30 cm layer compared with the deeper layers (Figure 2, I profile). The content of these sized particles in deeper layers markedly decreased. In our opinion, this must be related to the mineralization of Sapric Histosol in the 0–30 cm layer caused by his drainage and cultivation.

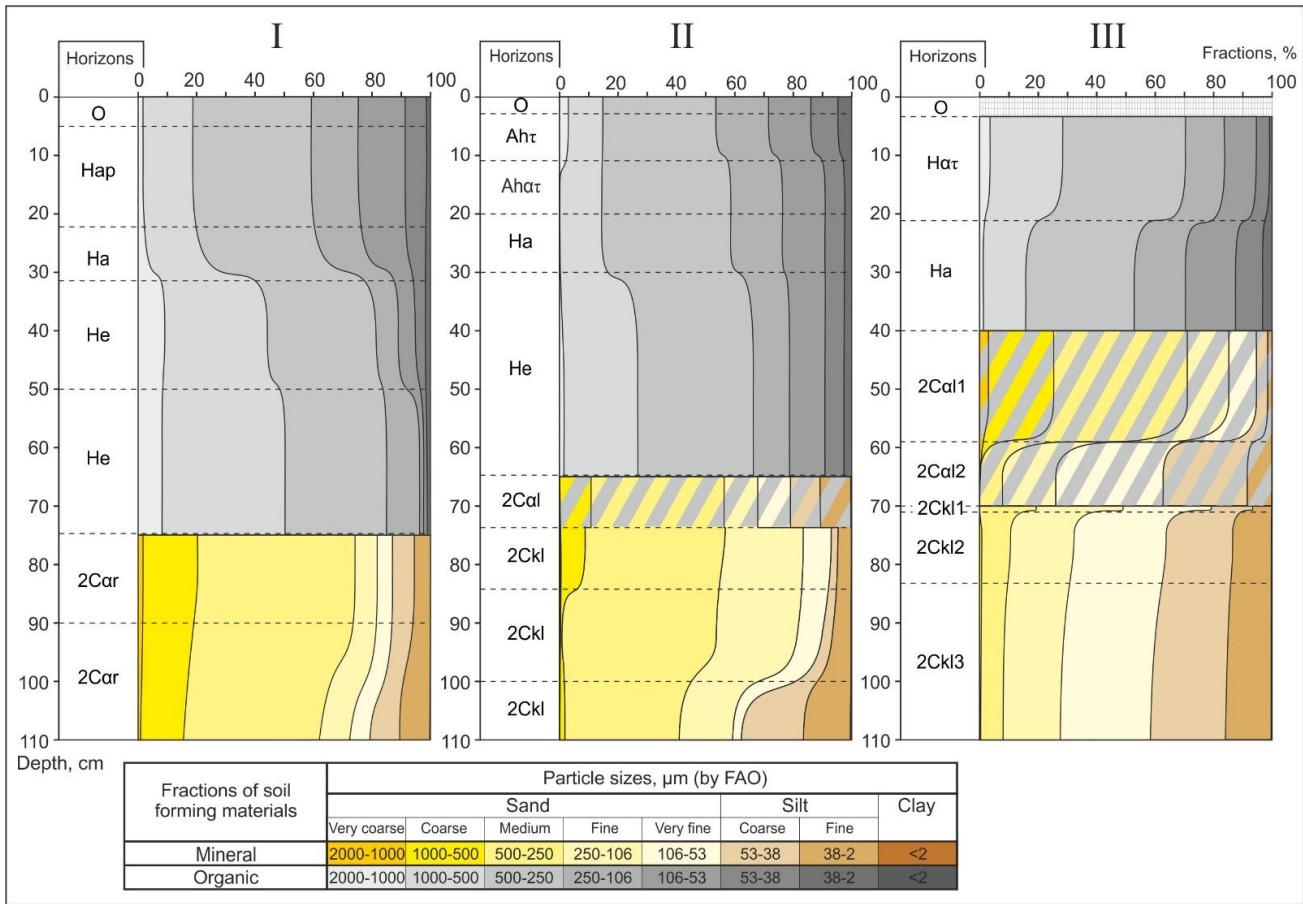

**Figure 2.** The textural composition of Sapric Murchic Histosol (**I**) and Sapric Darinic Histosol (**II,III**). Descriptions of master symbols and suffixes of horizons: O—forest litter or sod horizon with partly decomposed organic matter, A—a mineral horizon in which decomposed organic material is being accumulated, H—peat layer, C—initial mineral horizon, a—organic material in an advanced state of decomposition, e—organic material in an intermediate state of decomposition, h—humic horizon with significant amount of organic matter, i—organic material in an initial state of decomposition, k—secondary carbonates, l—capillary fringe mottling (gleying), p—horizon which is modified by cultivation (murshic), r—strong reduction conditions dominate, 2—a mineral horizon of another origin, α—primary carbonates, τ—human-transported natural material.

The texture in the organic part of the profile of Sapric Histosol with a removed peat layer was set the same as in the first profile. The content of particles whose size is 250–38 μm increased with the mineralization of peat horizons. The highest mineralization rate of the peat was established in the 0–30 cm peat layer (Figure 2, II profile). The ground surface of forest Sapric Histosol (Figure 2, III profile) is composed of limnoglacial sand, whose texture varies. The content of particles of the sand, whose fractions size are 500–250 μm and 250–100 μm, are decreasing, and the content of 53–2 μm (silt) particles is negligibly increasing. Sands whose origins are limnoglacial, at the 40–70 cm depth, are underlain by carbonated sapropel. Aquifuge is formed of them, which creates conducive conditions for peat layer formation. With the peat layer being removed in this Histosol, currently, only 37 cm of the peat layer remains. Him consists of strongly decomposed (Ha) and mineralized/mixed (Haτ) peats. The texture of the soil-forming ground surface of this Histosol is heterogeneous. This mineral layer consists of horizons that differ in genesis and texture. There are 2Ckl–2Ckr horizons present the sand layer, which are at 84–120 cm depth and formed at a deeper place of the limnoglacial basin under the effect of less intensive movement of water mass. This is evidenced by an increase in 53–2 μm particle fraction content. With decreasing thickness of the water mass, its movement affected the forming deposits: sand (500–53 μm) content increased while silt (53–2 μm) content decreased. The layer of the profile, which is at 70–72 cm depth, is particularly evident in this effect. Sapropel started to form after the basin had drained. More decomposed organic matter and 106–38 μm size particles dominate in sapropel present at 60–70 cm depth. Higher decomposed organic matter and 250–106 μm size particles dominate in sapropel at 40–60 cm. Content of 1000–250 μm size particles which are in the peat horizons increasing being effect of mineralization. In the analyzed profile, peat mineralization decreases with depth.

Fluctuations in pH values in the vertical profile are closely related both to the carbonate soil environment (presence of 2Ckr horizons) and to the cultivation of this soil. Therefore, in the 0–20 cm layer, we observe the increase in pH values, which should be related to the cultivation and mineralization of this layer, and at the same time, a decrease in pH values in the middle part of the profile where the natural moderately mineralized peat is present. From a depth of 50 cm, the pH values increase again (Figure 3. I profile). This is due to the carbonate soil environment from which the moisture lifts the dissolved carbonates through the capillary of the peat layer. As a result, the pH rises to 6.3 in the lower part of the profile.

The pH is very different in sward Sapric Histosol (profile II) because the profile of this soil is very heterogeneous in its origin, evolving not organically due to long-term and slow changes in the environment but due to the effects of intense and relatively short-term human anthropogenic activities. Fluctuations in pH values were mainly caused by the formation of artificial horizons (Ahτ—Ahατ) during peat excavation in this profile, in which the ratio of humic and mineral substances changes very strongly. The horizon has a higher pH and reaches 7 with a higher amount of humus, while on the horizon (Ahατ), the pH decreases to 6, most of which is sandy sediments. The pH value (~6.5) reflects the background value characteristic of this soil at depths deeper than 30 cm.

The trend of vertical distribution of pH of Sapric Histosol profile 3 (0–40 cm) is close to the first profile I (50–75 cm) (Figure 3). The depths differ because the analyzed peat is excavated. The pH of the 0–20 cm (Ha) horizon reflects the value of the natural peat of this Histosol, which, compared to profile I, is lower. This is because when this peat was mined, it was left to renaturalize on its own. The formed O horizon protected the Ha horizon from mineralization (Figure 3. profile III), and when it was not used in agriculture and was not fertilized, the pH remained unchanged compared to profile I. The pH of the Ha horizon at a depth of 20–40 cm is increased due to the influence of deeper carbonate horizons (2Cαl1—2Cαl2) and carbonated groundwater.

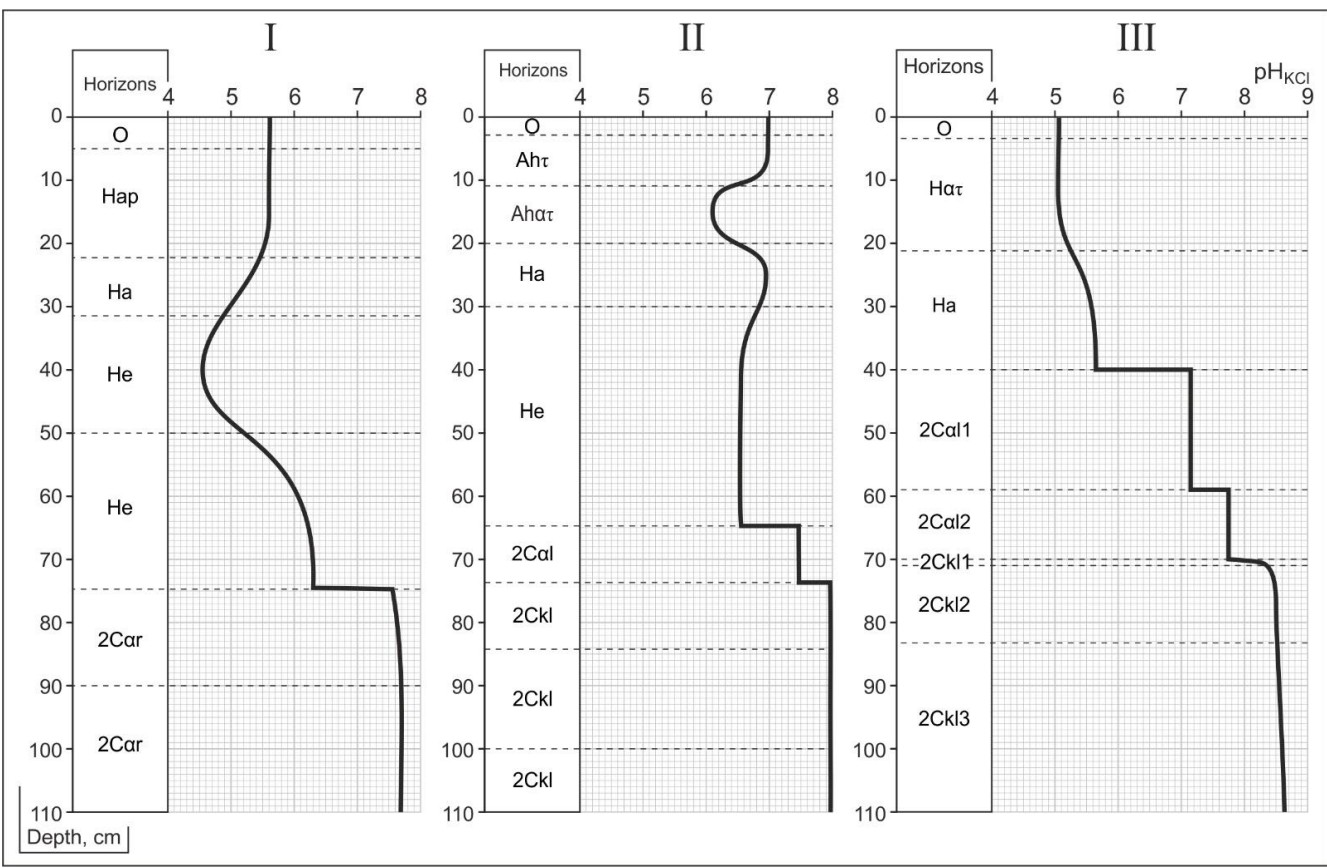

**Figure 3.** pH in the profile of Sapric Murchic (**I**) and Darinic Histosol (**II**,**III**).

The variation of the content of humus substances and degree of peat mineralization was affected by the content of the nitrogen (N) (Figure 4. I profile). As the mineralization of the peat material increased and the amounts of humus substances increased, the nitrogen content also increased. This is evidenced by the accumulation of nitrogen in the deeper layers of the organic part of the analyzed Sapric Histosol profile. The variation of phosphorus amount in the profile also is related to the peculiarities of histosol use.

The nitrogen content of the second histosol profile (Figure 4. II profile) varied in proportion to the amount of organic matter. Lower content of N was accumulated in the topsoil (up to 30 cm depth) of *Histosols* compared to in a layer deeper than 30 cm in which nitrogen content increased. Groundwater level fluctuations and lithogenic basis, which form the geochemical and geophysical barriers, are the main factors that determine the variation of phosphorus amount in this Histosol profile.

These fluctuations in values of phosphorus and potassium also had an effect on former fertilization and tillage. Similar variations of the values we determined in the first Sapric Histosol profile. N, P, and K values of profile III are analyzed in the context of profiles I and II because this Sapric Histosol, although excavated, was not used for agriculture (as profile I) and was not reclaimed (as profile II), i.e., part of its Hap-Ha profile had the opportunity to preserve its chemical properties relatively unchanged. Figure 4 shows that in profile III, N values fluctuate around $18$–$22$ g kg$^{-1}$. This correlates with the profile I $55$–$75$ cm He and profile II $30$–$65$ He horizons. These are relatively natural peat horizons that have not been affected either by fertilization during agricultural activities or reclamation by filling with mineral soil with low humus content. The amounts of K and P in profile III are minimal (K—$0.12$–$0.03$ g kg$^{-1}$, P—$0.00$–$0.00$ g kg$^{-1}$), like in profile I. This is explained by the fact that this Sapric Histosol is not used in agriculture and is not fertilized. A slight increase in the amount of K in the $0$–$20$ cm layer is associated with the leaching of K and the O horizon formed by the fall of tree leaves and relative accumulation in the $5$–$20$ cm Ha$\tau$ peat horizon.

Larger accumulations of N, P, and K at a depth of 40–70 cm are related to the geochemical sorption barrier formed by carbonated organomineral deposits (sapropel).

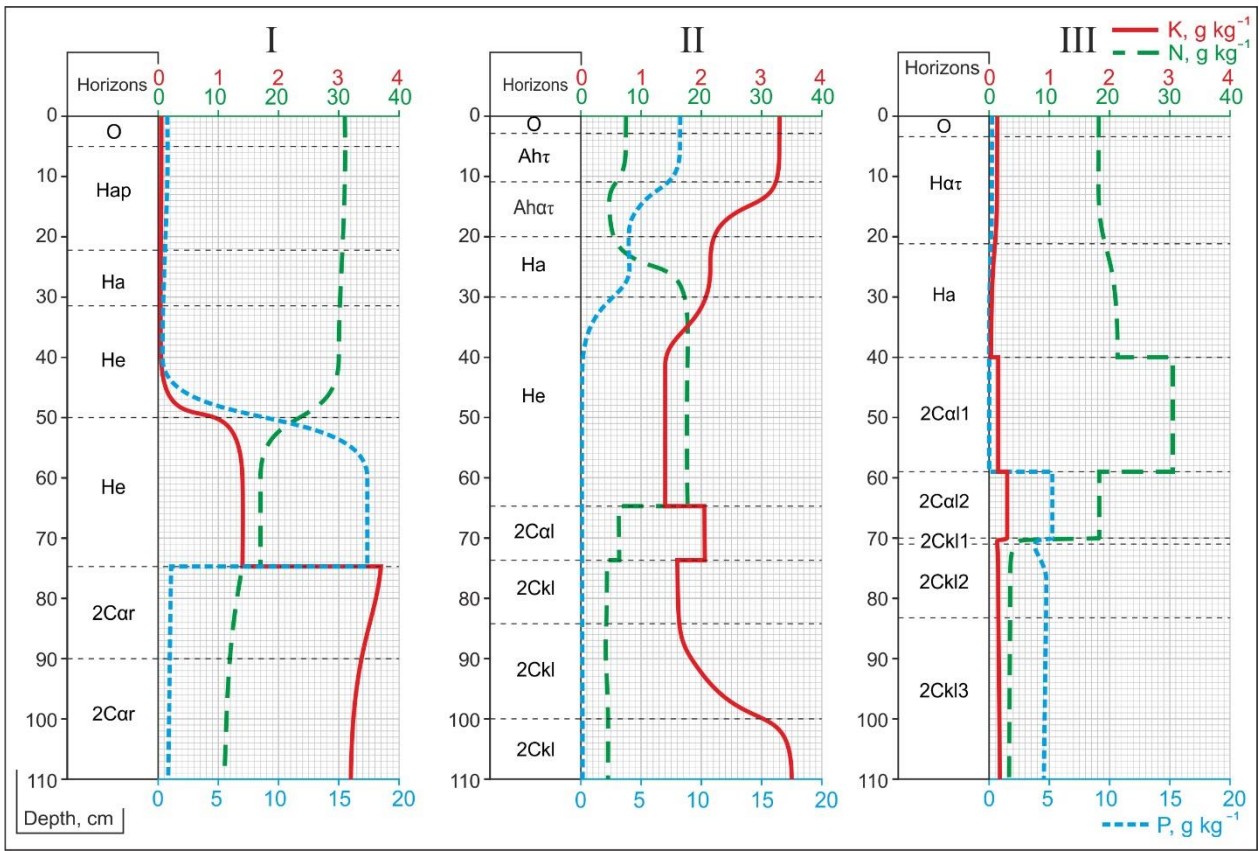

**Figure 4.** Distribution of the amounts of macroelements (N, P, K) in the profile of Sapric Murchic Histosol (**I**), and Sapric Drainic Histosol (**II,III**).

The peculiarities of distribution of SOC and DOC in the profile of Sapric Histosol are related to the mineralization of peat material (Figure 5). This leads to the fact that SOC content (0–30 cm) decreased, while DOC content increased in the upper 0–30 cm layer of Histosol (I profile). The pronounced limit of peat mineralization passes at a depth of 30 cm, and this is also evident from the distribution of physical particle size data (Figure 2): the concentration of DOC is considerably reduced, and SOC content recovered at this depth, respectively.

In the second profile, SOC content decreased, while DOC content increased in the 0–30 cm layer of Sapric Histosol. The SOC content increased, and DOC content decreased in the natural organic part of the profile (30–65 cm) exposed to anthropogenic activities (Figure 5, II profile). In profile III, the SOC content is twice as low as in the peat horizons of the other investigated Sapric Histosol profiles. This should be explained in the genesis of the peat and the time of decomposition. In the profile, the variations in the peat horizons are small and are associated with a relatively higher degree of mineralization in the 0–20 cm top layer. DOC variations in profile III are directly related to C concentration and do not reflect analyzed Sapric Histosol use or cultivation characteristics.

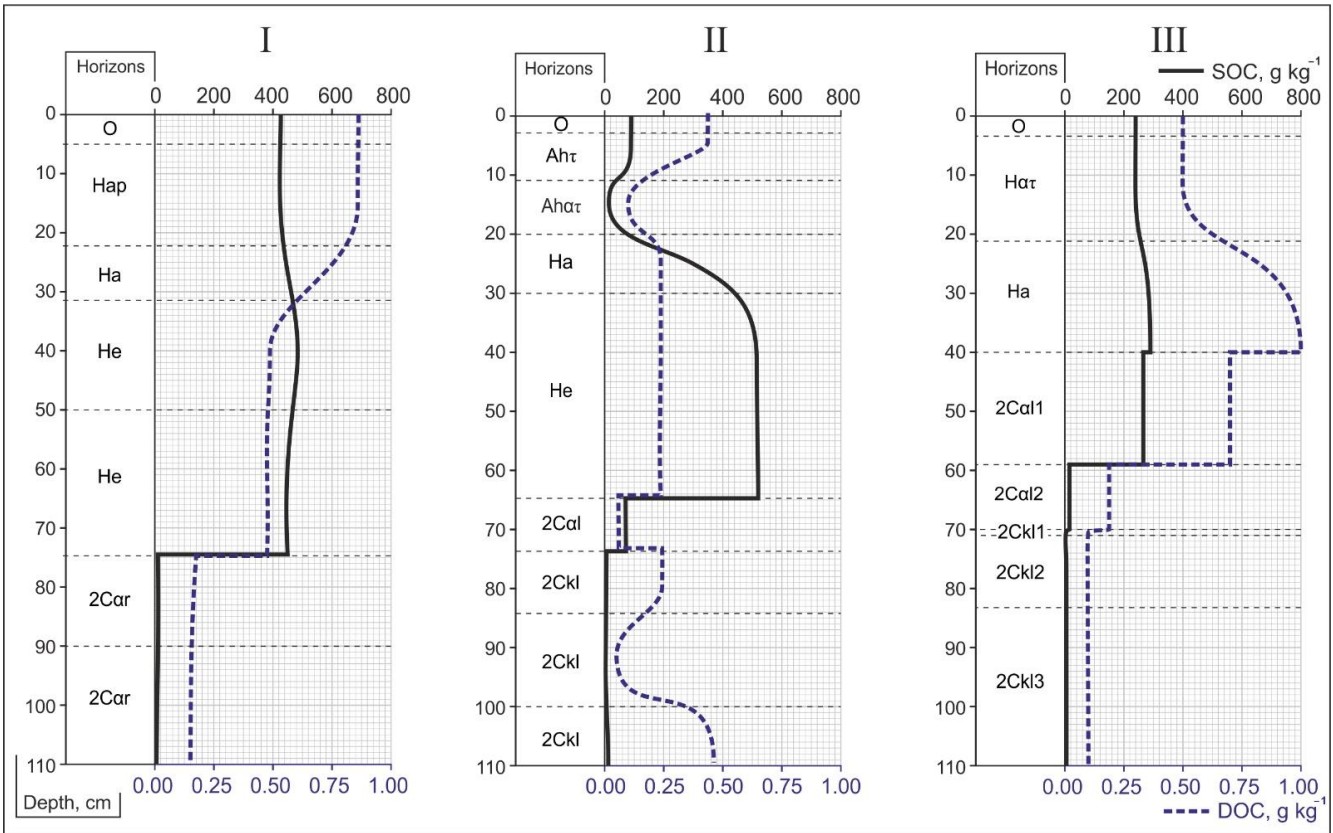

**Figure 5.** Distribution of the amount of soil organic carbon (SOC) and dissolved organic carbon (DOC) in the profile of Sapric Murchic Histosol (**I**) and Sapric Drainic Histosol (**II,III**).

The amount of mobile humus substance (MHS) in the soil is directly related to the processes of humification and mineralization of organic matter in the soil, i.e., processes of pedogenesis taking place in the peat layer of the Histosol. These processes are strongly dependent on human economic activity (Figure 6). The distribution of MHS and MHA (Figure 6, I profile) in the Sapric Histosol profile indicates a direct impact of land use on the development of Sapric Histosol. Due to this, the quantity of MHS has decreased in deeper than 30 cm *Histosols* layers. The more decomposed the peat, the more MHS it contains, so the amount of MHS in the He (moderately decomposed peat) horizons ranges from 60–70 g kg$^{-1}$, and in Ha (highly decomposed peat)—85–95 g kg$^{-1}$. Atmospheric effects and intensive humification processes in the topsoil of Sapric Histosol make MHS content close to 150 g kg$^{-1}$ (profile III), while intensive conventional tillage in Sapric Histosol promotes its mineralization and, therefore, MHS content decreases (profile I). Profile II stands out based on its results. Since its top layers are formed in the recultivation process by adding a mixture of mineral matter and humus, the MHS content in it exceeds 500 g kg$^{-1}$. The ratio between MHS and MHA shows the effect of human economic activities on Histosol. Intensive plowing and fertilizing (profile II) drastically reduce the amount of MHA and, at the same time, reduces the durability of MHS. Cultivation of perennial grasses (profile I) increases the amount of MHA (profile I), meantime spontaneous renaturalization and the growth of hardwood deciduous tree forests to the formation and accumulation of mobile fulvic acid (MFA) in the topsoil of Sapric Histosol (profile III).

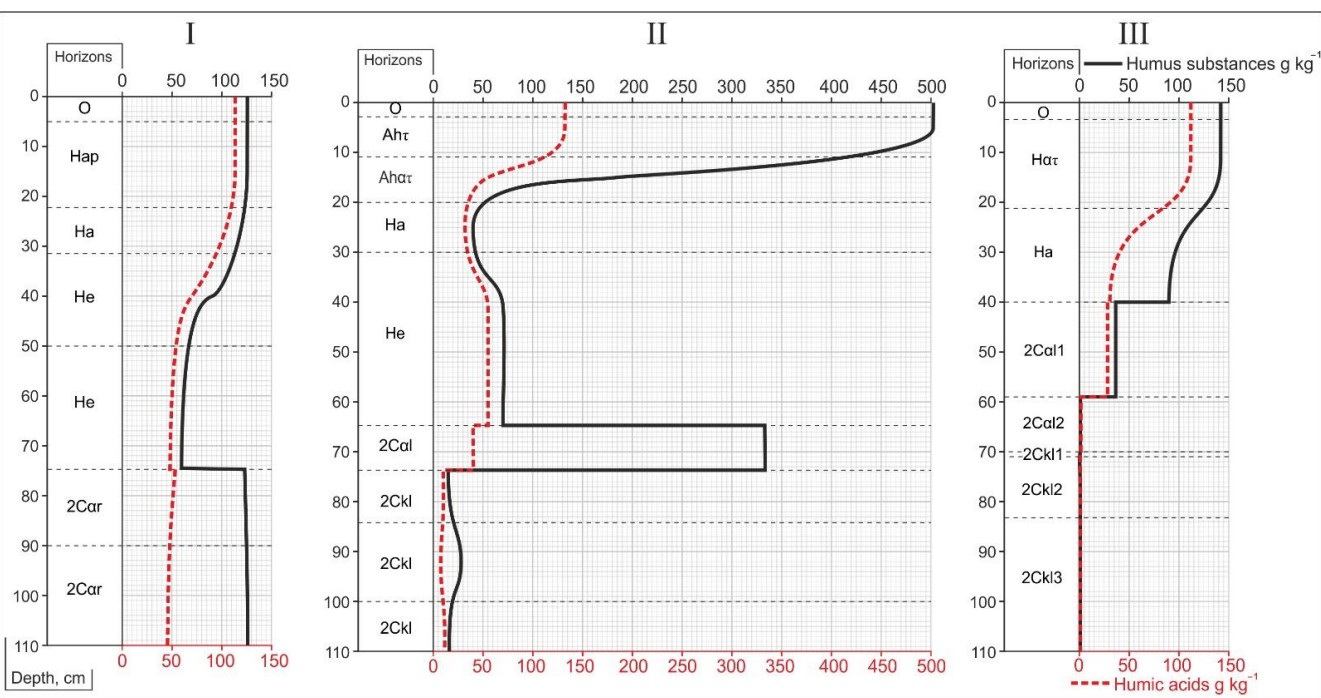

**Figure 6.** Distribution of the amounts of humus substances in the profile of Sapric Murchic Histosol (**I**) and Sapric Drainic Histosol (**II,III**).

### 3.2. Physical Properties of Sapric Histosol

Different Sapric Histosols were used and removed from the peat layer, affecting its different physical properties: bulk density, moisture, and total porosity. The total porosity of analyzed histosols is directly related to the bulk density, given that the higher the bulk density, the lower the total porosity and vice versa in the peatland. The lowest bulk density and, at the same time, the highest moisture and total porosity were established in the previously fertilized perennial grasses compared with other treatments of Sapric Histosol with a non-removed peat layer. The results of bulk density and total porosity, which are measured in Sapric Histosol with removed peat layer, showed that the lowest bulk density and the highest total porosity were established in the Sapric Histosols, which are under natural forest. Further development and renaturalization of this Histosol result in the accumulation of weakly degraded and less mineralized organic matter compared to the sward or crop rotation. The highest value of bulk density and the lowest value of total porosity and moisture were determined in the Sapric Histosol with removed peat layer, which are under the sward (Table 2).

**Table 2.** The effect of land use on physical properties in Sapric Histosol.

|  | Bulk Density g cm$^{-3}$ | Moisture m$^3$ m$^{-3}$ | Total Porosity % |
|---|---|---|---|
| Sapric Murshic Histosol previously used under the effect of renaturalization | | | |
| non-used | 0.257 abY | 2.13 bYZ | 90.2 abX |
| unfertilized perennial grasses | 0.285 aY | 2.27 bYZ | 89.2 bX |
| fertilized perennial grasses | 0.237 bY | 3.00 aX | 91.0 aX |
| red clover and timothy mixture | 0.283 aY | 2.20 bYZ | 89.3 bX |
| crop rotation field | 0.269 abY | 2.36 bXYZ | 89.8 abX |
| Sapric Drainic Histosol with no peat layer in use | | | |
| natural forest | 0.184 bY | 1.89 bZ | 93.0 aX |
| crop rotation field | 0.211 bY | 2.78 aXY | 92.0 aX |
| sward | 0.574 aX | 1.04 cE | 78.2 aY |

Note. The different letters a–c in the column indicate significant differences ($p < 0.05$) in the parameters of peat soil among different treatments in Murchic or Drainic Histosol, and the different letters X, Y, Z, and E indicate significant differences ($p < 0.05$) in parameters of all treatments.

### 3.3. The Effect of Sapric Histosol Land Use on Qualitative Changes in pH, Soil Organic Carbon, Dissolved Organic Carbon, and Humus Substance

No significant pH differences were observed in the upper layer (0–30 cm) of the histosol profile, which are typically in renaturalized peat regardless of the previous use. This may have been due to drainage, and previous use (perennial and bean perennial grasses) has resulted in the formation of the same pH during Sapric Histosol use (Table 3). Leaching of organic acids and the formation of a layer of sediments promotes the acidification of natural forest Sapric Histosol, as well as an increase in acidification with continuous and intensive tillage. At the same time, crop rotation promotes crop-friendly pH. Maintenance of perennial grassland promotes the accumulation of organic matter, slows down the mineralization of peat, and forms a neutral pH of Sapric Histosol.

**Table 3.** The effect of renaturalization and land use on pH, SOC, DOC, mobile humus substances (MHS), mobile humic acids (MHA), mobile fulvic acids (MFA), and humic acids to fulvic acids ratio (MHA:MFA) in Sapric Histosol.

| | pH$_{KCl}$ | SOC | DOC | MHS | MHA | MFA | MHA:MFA | HD | E4:E6 |
|---|---|---|---|---|---|---|---|---|---|
| | Sapric Murshic Histosol previously used under the effect of renaturalization | | | | | | | | |
| | g kg$^{-1}$ | | | | | | | | |
| non–used | 6.0 aZ | 408 bX | 1.02 aX | 178 aX | 75.6 bY | 102 aX | 0.74 aE | 18.5 bY | 3.96 aYZ |
| unfertilized perennial grasses | 6.0 aZ | 409 bX | 0.95 aX | 186 aX | 85.9 aX | 100 aX | 0.86 aE | 21.0 aX | 3.88 aYZ |
| fertilized perennial grasses | 6.0 aZ | 435 aX | 0.91 aX | 188 aX | 86.5 aX | 101 aX | 0.85 aE | 19.9 abY | 4.06 aYZ |
| red clover and timothy mixture | 6.0 aZ | 416 bX | 0.92 aX | 189 aX | 85.3 aX | 103 aX | 0.83 aE | 20.5 aX | 4.00 aYZ |
| crop rotation field | 6.1 aZ | 418 abX | 0.92 aX | 182 aX | 82.7 aX | 99 aX | 0.83 aE | 19.8 abY | 3.87 aYZ |
| | Sapric Drainic Histosol with no peat layer in use | | | | | | | | |
| natural forest | 5.9 cZ | 343 aY | 0.95 aX | 80.8 aY | 65.3 aZ | 15.5 aY | 4.22 bY | 19.0 aY | 4.24 bY |
| crop rotation field | 6.6 bY | 180 bZ | 0.48 bY | 25.2 bZ | 16.4 bE | 8.78 bZ | 1.87 cZ | 9.12 bZ | 3.66 cZE |
| sward | 7.3 aX | 64 cE | 0.39 bY | 8.9 cE | 8.05 cF | 0.90 bE | 8.97 aX | 12.5 bZ | 5.80 aX |

Note. The different letters a–c in the column indicate significant differences ($p < 0.05$) in the parameters of peat soil among different treatments in Murchic or Drainic Histosol, and the different letters X, Y, Z, E, and F indicate significant differences ($p < 0.05$) in parameters of all treatments.

SOC content relates to the morphology and land use of Sapric Histosol. The highest amounts of soil organic carbon were established in Sapric Histosol with a non-removed peat layer, compared with Sapric Histosol with a removed peat layer. It was caused by the fact that after peat layer removal, the top layer of the peat soil profile was mixed with the mineral matter, which is transported, as a result of that reduction of organic carbon content. An exception is Sapric Histosol which is with a removed peat layer and is under natural forest, where organic matter increases with the natural accumulation of organic matter on the forest floor. Variations of SOC content are affected using histosol and intensive mineralization of Sapric Histosol with a non-removed peat layer. Unused Sapric Histosol accumulated less SOC because of the ongoing processes of histosol mineralization, and former perennial grasses fertilized with NPK provide conditions for the accumulation of a higher amount of SOC. The results for dissolved carbon show the same regularity as for total SOC. The amount of DOC varies from 0.91 to 1.02 g kg$^{-1}$ in Sapric Histosol with a non-removed peat layer and 0.39–0.95 g kg$^{-1}$ in Sapric Histosol with a removed peat layer (Table 3). Due to analyzed Histosol cultivation and His peat material mineralization, form organo-mineral-compounds, which are less mobile and soluble and decrease soil DOC content, include mobile organic acids. In the Histosols that are under natural forest, were established higher (0.95 g kg$^{-1}$) amounts of DOC compared with other treatments in Sapric Histosol with a removed peat layer. It is related to total carbon accumulation on the forest floor. The lowest amounts of DOC were found in the sward and crop rotation field. Removed of the peat layer and the previous cultivation of perennial grasses and now currently renaturalization peatlands directly determined the higher amounts of mobile humus substances and mobile humic acids (MHS 186–189 gkg$^{-1}$ and MHA 85.3–86.5 gkg$^{-1}$). For the opposite, non-used

Sapric Histosol, after draining, affected the lowest accumulation of MHS and MHA in the Sapric Histosol with a non-removed peat layer. Forest Sapric Histosol, due to natural conditions: fallows and moisture, accumulates more humus substances and humic acid compared to cultivated fields and sward. Despite this, it has a high MHA:MFA ratio. It should not be associated with negative mineralization processes. This is likely due to the stability of humic acids and the mobility and leaching of fulvic acids. Thus, the pH under deciduous hardwood forests in the investigated object is more acidic (pH 5.1) compared to other research objects. The removal of the peat layer and agricultural activities resulted in the lowest accumulation of MHS and MHA in sward in Sapric Histosol with a removed peat layer.

Better quality of the peat soil in terms of its MHS:MFA ratio was found in the renaturalized fields than in the Sapric Histosol with removed peat layer (Table 3). The previous cultivation of perennial grasses (MHA:MFA 0.83–0.86) had a positive affected on the ratio of humic and fulvic acids compared to unused peat soil (MHA:MFA 0.74). The MHA:MFA in forest, crop rotation, and meadow Sapric Histosol is above 1, so it is determined that a loss of labile fractions is taking place, i.e., mineralization of peat materials.

Sustainable and climate change-neutral use of Sapric Histosol is associated with humification processes occurring in the soil. When HD > 20—humification takes place, and when HD < 20—mineralization begins to appear. According to the degree of humification, it was set that secondary humification is faster in renaturalized fields and forest Sapric Histosol compared to other research treatments. Humification was slowest in non-used Sapric Histosol with non-removed peat layer and in the fields of sward and crop rotation, where was Sapric Histosol with removed peat layer.

According to the results of the study, sowing the field with perennial and legume grasses is the most favorable in terms of organic carbon stabilization in drained Sapric Histosol. A significant shift in the MHA:MFA ratio in favor of humic acids is characteristic of all affected by renaturalization Sapric Histosol study variants with non-removed peat layer and sward compared with Sapric Histosol used for agriculture. Previous cultivation of perennial grasses, such as their fertilization or forest implantation, has a positive effect on the humification of humus substances.

*3.4. Spectral Properties of Humic Acids of Sapric Histosol*

The ratio of humic acids (E4:E6 ratio) and optical density is one of the indicators of humic acids that indicates the quality of humic acids and processes of humification in the soil. The conducted research determined the high values of the E4:E6 ratio. It indicates that humic acids are characterized by low aromaticity and large quantities of aliphatic structures, carbohydrates, and amides in humic acids. When E4:E6 is <4, it indicates high polymerization of humus substances and, according to stability, low mobility and high quality. According to our data, low aromaticity humic acids and the highest concentration of aliphatic structures were established in sward Sapric Histosol. In contrast, the highest aromaticity humic acids were identified in the crop rotation field (Table 3). Similar E4:E6 values (3.87–4.06) were determined in the renaturalized Sapric Histosol. Fertilization of perennial grasses and cultivation of legume grass resulted in higher E4:E6 values than in the previous crop rotation field, unfertilized perennial grasses field, or unused Sapric Histosol. The highest value of optical density (E4:E6) (4.06) was established in the previously fertilized perennial grasses and in the red clover-timothy mixture. Conditions including non-used, unfertilized, and cultivation status affected the highest aromaticity of humic acids (Table 3).

*3.5. The Effect of Previously Used on Molecular Indicators of Dissolved Organic Matter*

HPLC-SEC analysis was performed to identify the chemical composition and molecular parameters of humus substances. Molecular indicators of organic substances show their composition and which molecules predominate in their composition.

DOC, MHS (MHA and MFA), proteins, and low molecular mass classes of compounds were found in the Sapric Histosol (Figure 7). Previously, conventional tillage and fertilization affected differences in Sapric Histosol organic matter at the molecular level. Homogeneous organic matter was found in the Sapric Histosol variant with a non-removed peat layer. DOM is dominated in all Sapric Histosol use. Low molecular weight compounds were set in former crop rotation fields and unfertilized perennial grasses, while the protein in other variants in Sapric Histosol with a non-removed peat layer. The molecular composition of Sapric Histosol with a removed peat layer was found to be different. DOM, protein, and mobile humic acids were set in forest Sapric Histosol. Fulvic and humic mobile acids, low molecular weight, and DOM were found in sward and crop rotation fields (Figure 7).

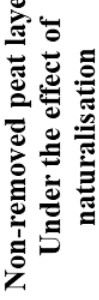
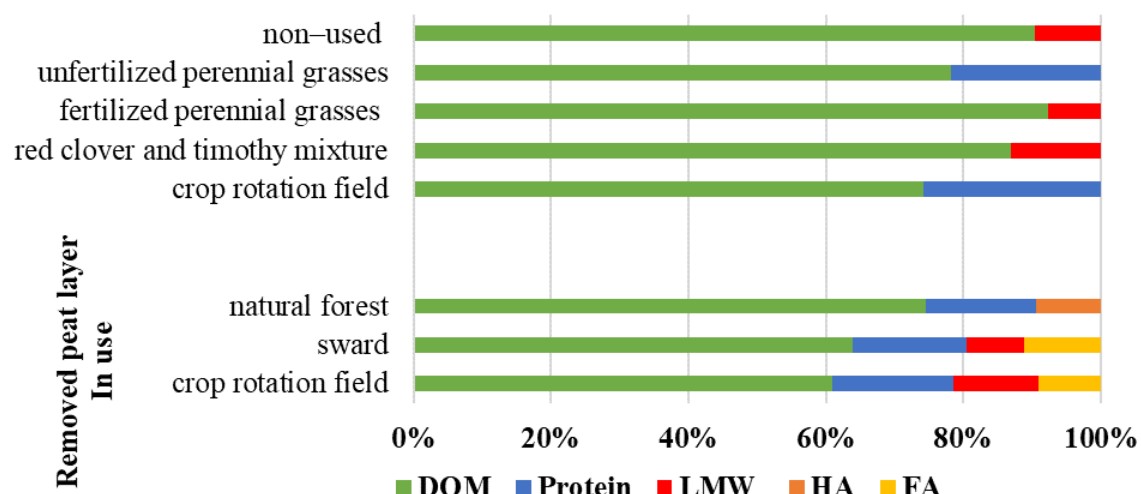

**Figure 7.** Water extractable DOM molecular fractions of Sapric Murchic Histosol with a non–removed peat layer and Sapric Drainic Histosol with a removed peat layer by HPLC (%). Note: DOM—dissolved organic matter, LMW—low molecular weight compounds, HA—humic acids, FA—fulvic acids.

According to the results (Figure 7), DOM with larger molecular weight (Mw) contains more aliphatic C, and the aliphatic C chains have lower contents of C=C double bonds. This might be the result of the degradation of labile organic matter constituents under the action of microbes.

Differences in soil organic matter at the molecular level were still found even 14 years (since 2001) after the abandonment of Sapric Histosol use for agricultural purposes. Average molecular weight (Mw) and polydispersity index were calculated, examining the impact of past Sapric Histosol used on the organic matter at the molecular level (Table 4). As the molecular weight increases, the potential for material degradation increases. The highest molecular weight compounds of dissolved organic matter were found in the renaturalized crop rotation (molecular weight (Mw)—6886 Da) and red clover-timothy field (Mn—4937 Da). This indicates that these soils are dominated by unstable water-soluble carbon compounds, which can be rapidly broken down by microorganisms and removed from the soil. Lower molecular weight organic matter (unused and unfertilized perennial grasses) has less mobility and is more absorbed by mineral surfaces.

**Table 4.** Molecular characteristics of dissolved organic matter of Sapric Histosol.

| | Mean of Number (Mn), Da | Mean of Weight (Mw), Da | Polydispersity Index (PDI) |
|---|---|---|---|
| Sapric Murshic Histosol previously used under the effect of renaturalization | | | |
| non–used | 1818cE | 2207cE | 1.215aXY |
| unfertilized perennial grasses | 1907cE | 2334cE | 1.224aX |
| fertilized perennial grasses | 2847bcZ | 3282bcZ | 1.162bcZE |
| red clover and timothy mixture | 4368bY | 4937bY | 1.135cE |
| crop rotation field | 5960aX | 6886aX | 1.155bcZE |
| Sapric Drainic Histosol with no peat layer in use | | | |
| natural forest | 1819bE | 2196bE | 1.208aXY |
| crop rotation field | 3010aZ | 3551aZ | 1.180cYZ |
| sward | 1757bE | 2096bE | 1.193bY |

Note. The different letters a–c in the column indicate significant differences ($p < 0.05$) in the parameters of peat soil among different treatments in Murchic or Drainic Histosol, and the different letters X, Y, Z, E indicate significant differences ($p < 0.05$) in parameters of all treatments.

Low molecular weight organic matter (Mn 3551–2096 Da) dominated Sapric Histosol with a removed peat layer. The lower molecular weight organic matter fractions are characterized by higher aromaticity (low molecular weight aromatic compounds with more C=C double bonds dominate) and decreased mobility, resulting in increased humification and stability of humus substances. Additionally, it could be used to improve carbon sequestration and soil quality. The polydispersity index of dissolved organic matter (homogeneity of organic matter) ranges from 1.135 to 1.224. Non-used, unfertilized perennial grasses in Sapric Murshic Histosol, natural forest, and sward peat soil used affected to increase unprotected forms from mineralization. Perennial grasses fertilizing and Sapric Histosol use for legume grasses determined stable and homogeneous organic matter.

During the 14-year renaturalization process (since 2001), changes are still observed not only in the peat soil's chemical composition and physical properties but also in the molecular level of organic matter.

Molecular studies of dissolved organic matter by high-performance chromatography have shown that higher molecular weights are characteristic of Sapric Histosol under renaturalization than Sapric Drainic Histosol.

*3.6. Hydrophobicity by FT–IR Absorption Spectra of Sapric Histosol*

Analysis of humic acids by FT–IR absorption spectra showed that different land-use/non-use and renaturalization of Sapric Histosol affected the qualitative parameters of humic acids. All spectra show some differences indicating changes in the chemical composition of the sample (Figure 8). Humic acids are composed of several chemical groups, some of them having quite similar absorption frequencies. Thus, it is complicated to have a very detailed assignment of spectral bands. However, some very typical chemical groups can be easily identified. One of the easily identifiable absorption bands is a part of the doublet, with the peak at 1718 cm$^{-1}$ readily assigned to the C=O stretch vibrations of the carboxyl group (-COOH) in humic acids. Another part of this doublet centered at 1637 cm$^{-1}$ associated with C=C and C=O stretch vibrations, mostly from the ketonic groups. Another typical band is a very broad feature between 3600–2200 cm$^{-1}$; this band arises from various O-H stretch vibrations. The broad nature of this band arises from the possibilities to form various hydrogen bonds. Band group between 3000–2700 cm$^{-1}$ is assigned to C-H stretch vibrations from $CH_3$, $CH_2$, and CH groups. Assignment of the bands in the 1400–600 cm$^{-1}$ region is not that straightforward, a rise from stretch and deformational vibrations from C-O, C-N as well as various deformation vibrations of CH groups (Figure 8.).

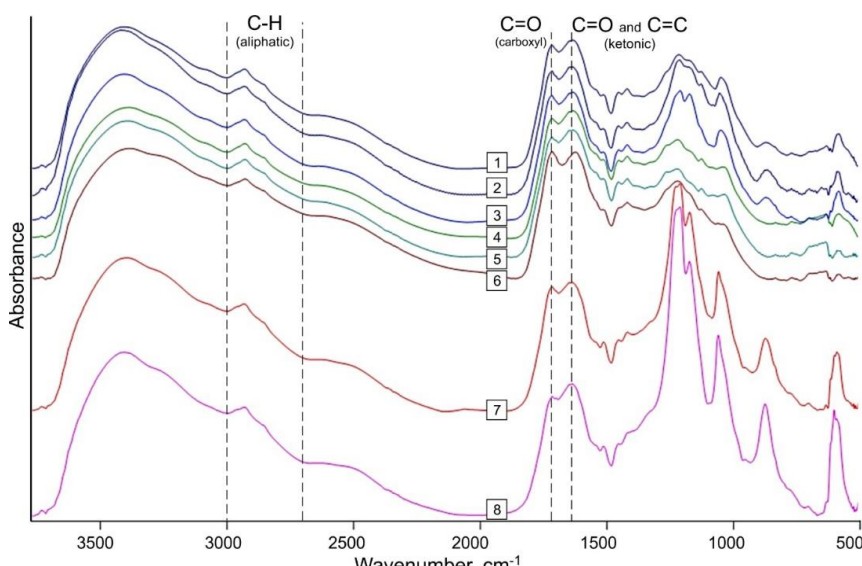

**Figure 8.** Infrared absorption spectra of soil samples. Spectra are shifted in Y axis for the sake of clarity. Sapric Murchic Histosol: 1—non-used peat soil, 2—unfertilized perennial grasses, 3—fertilized perennial grasses, 4—red clover and timothy mixture, 5—crop rotation field; Sapric Drainic Histosol: 6—natural forest, 7—crop rotation field, 8—sward.

Based on previous attempts to characterize the hydrophilicity of the soil samples [25,32], the ratio between the bands of highly hydrophobic C-H groups and hydrophilic C=O groups could be used as a qualitative measure of the sample hydrophilicity [30]. We used an integrated intensity of the O-H stretch region and integral intensity of both C=O stretch bands to estimate the hydrophilicity of the samples. The highest ratio of the C=O to O-H band integrals, thus the highest concentrations of hydrophilic groups as well, was found in forest Sapric Histosol as indicated in Figure 9. The ratio values for samples in renaturalizable fields previously fertilized perennial grasses, legumes, and crop rotation fields were like each other and showed slightly lower concentrations of hydrophilic groups. The lowest hydrophilicity is of the crop rotation Sapric Histosol with removed peat layer. While sward, previously unfertilized perennial grasses, and non-used Sapric Histosol had a similar, only slightly increased hydrophilicity as that of crop rotation field Sapric Histosol.

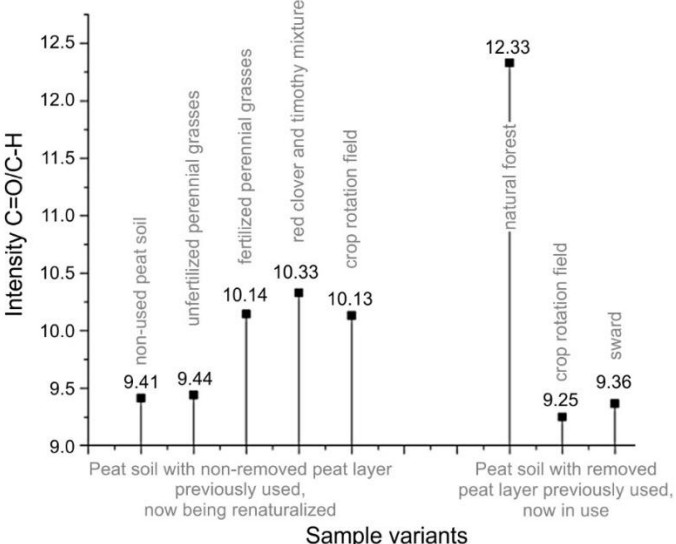

**Figure 9.** Intensity ratio of the integrated absorption of C=O stretch bands (1718 and 1637 cm$^{-1}$) and C-H stretch bands (range 3000–2700 cm$^{-1}$).

## 4. Discussion

Discussions of peatlands used in agriculture are becoming very relevant and especially related to the European Green Deal [17,28]. It is possible to talk about non-use only of those peatlands that are natural and not affected by land reclamation. After draining peatlands, we must choose one or another way of using them. Objective research data allow choosing the type of peat land use that would least promote the mineralization of SOM and peat material and preserve not only its quantity but also its quality, correspondingly reducing GHG emissions. Studies of Sapric Histosol profiles have shown that regardless of whether the Histosol is mining up or not, the optimal way of its use must be associated with the maintenance of a constant plant cover and the formation of a permanent bed (O horizon). The most reliable cover contributing to the stabilization of SOM and the promotion of humification processes is formed by perennial grasses with red clover, as well as the cultivation of deciduous forests. According to Montanarella and Panagos [17], agroforestry is one of the alternatives for economic activity in agricultural Histosols, especially—Sapric Histosols.

The amount and quality of humus substances and humic acids are indicators of Histosol quality. By evaluating the qualitative indicators of soil humus: the content of humic acids, HA:FA, HD, and the optical densities of humic acids (E4:E6), we can decide whether the soil is being used sustainably or is undergoing degradation.

The results obtained by us confirm and supplement the knowledge about the differences in the qualitative composition of humic acids during renaturalization or different use of peat after drainage and removal of the peat layer. A removed peat layer resulted in lower levels of SOC, DOC, and humus substances. Minimal differences were found in the renaturalized Sapric Histosol between former perennial grass cultivation and unused Sapric Histosol. The humification degree (HD), which is understood as the relative proportion of HA to SOC, shows the resistance of SOM to environmental factors. The increase in HD values is valuable in both an agronomical approach in the context of sustainability for soil use and an ecological approach in the context of ecosystem stability [32]. The highest degree of humification due to natural processes was determined in the renaturalized and forest Sapric Histosol compared to other investigation treatments. The MHA:MFA ratio is known as an indicator of the polymerization and condensation of SOM and shows the ratio of potential mobility of organic C in the Histosol. An MHA:MFA ratio close to one indicates a stabilization of mobile humus substances and good quality of SOM, and a ratio greater than one points out that the mobile fractions are lost and indicates degradation of SOM quality [32,34,50–52]. The renaturalized Sapric Histosol had a better quality of humus according to MHA:MFA than the other research variants, as the values are close to one. Forest, sward, and crop rotation Sapric Histosol values were more than one because mobile and labile fractions were lost.

According to the results of the study by Klavin and Purmal [30], higher HA E4:E6 ratios measured in deeper peat layers are characterized by a higher degree of decomposition and age of the peat, a lower degree of condensed aromatic systems and smaller particles sizes or molecular weight. Moreover, the concentration of carboxyl and phenolic hydroxyl groups and the increase in carboxylic acidity depend on the age of the peat and the degree of its decomposition, depth of measurements, and humification degree of peat materials. Sjadak [16], who compared the ratio of drained and undrained peat, found that E4:E6 tended to be higher in drained peat compared to E4:E6 in undrained peat. A lower degree of peat humification, aromatic condensation, and polyconjugated HA molecules, when compared to undrained peat soils, was found in drained peat soils. The E4:E6 ratio results of our study of 3.66–5.80 are like Mielnik [53]—E4:E6 values were in the range of 4.6–5.0.

The results of our study support the conclusions of other researchers. Bojko [54] and Kalisz [33] found in their research that the E4:E6 ratio of the humus acids was the lowest in peat soil which is characterized by a large number of particles of the silty fraction size, and higher, in unaltered peat soils or in peat soils with a slight amount of particles of the silty fraction. It indicates that secondary humification can be associated with an increase in the number of particles of the silt fraction size. It has been suggested that the low E4:E6 ratio

indicates the accumulation of stable forms of organic substances due to the higher part of humic acids. Due to secondary humification, the low values of the E4:E6 ratio suggest the prevalence of mature humus but not necessarily the most stable forms of SOM. As a result of secondary humification, SOC stored in the peat material of Histosol, as the most stable form of SOM, is transformed into fulvic and humic acids. The results of our study showed that the correlation between MHA:MFA and E4:E6 is 0.566 ($p < 0.05$). The MHA:MFA ratio and the E4:E6 ratio in the renaturalized Sapric Histosol with a non-removed peat layer in peat soil are lower than in Sapric Histosol with a removed peat layer in peat soil, which shows and confirms the conclusions of other researchers that stable humus substances are formed here. Moreover, in the Sapric Histosol with removed peat soil layer, in peat soil, there was a higher E4:E6 ratio and MHA:MFA above one, indicating HR instability. In perspective, it would be necessary to compare renaturalized peat with natural peat and to confirm Howson's [3] statement, which says, according to E4:E6, the blanket fen peat is more humified and aromatic than that of the raised bog.

Low molecular weight compounds indicate mineralization [5]. This is confirmed by the results of our research. Dissolved organic matter previously used for red cover mixture, crop rotations, and fertilized perennial grasses peat soil and now under renaturalization is more stable and protected from mineralization than other treatments. The most sensitive DOM to mineralization, according to the polydispersity index, is non-used, nonfertilized perennial grasses, forest, sward, and crop rotation Sapric Histosol. After evaluating the hydrophobicity of humic acids, we can decide about the mineralization and degradation of organic substances.

Szajdak [16] and Wilson [1] claim in their works that the hydrophilicity of peat's matter surface is associated with the availability of organic functional groups (phenolic, alcoholic hydroxyls, and carboxyls) capable of hydrogen bonding. These oxygenated structures are present in humic acids. The water retention properties of peat soils are related to the molecular configuration, structure, and origin of the peat material. Humic acids are thought to be mainly responsible for the hydrophilicity of peat. The number of hydroxyl and other polar bonds affects the hydrophilicity of humic acids. With the results of our study, which compares the hydrophilicity of humic acids in the renaturalized peat soil and peat soil with a removed peat layer currently used for forest, agriculture, and sward, we add to the knowledge of other researchers.

Research by Zaccone [55] shows that the aromatic/aliphatic ratio increases from the depth of 70 cm in the peat soil profile while the carboxyl group bonds decrease, as shown by the FT-IR spectra. Our study found that after draining the peat, it is best to use it for forest or for growing perennial grass. This is evidenced by the FT-IR analysis spectra, which show the least damaged and retained hydrophilicity.

## 5. Conclusions

Our research has shown that the most pronounced changes in Sapric Histosol, which is drained, renaturalized, and used differently, occur in its 0–30 cm top layer. It significantly increases the amount of 106–2 µm particles, pH value, and labile organic carbon compared to deeper layers. This testifies to the mineralization of the peat material taking place in this layer. Peat mineralization is slowed down by the cultivation of deciduous hardwood forests and the formation of forest litter (O horizon). This is shown by the increase in the number of particles in the 1000–250 µm interval in topsoil under the O horizon of the investigated Sapric Histosol profile.

The same peculiarities are present in all land uses of Sapric Histosol, in which bulk density increases with depth. The bulk density of 0.24 g cm$^{-3}$, which is the lowest of the studied variants, was found in the former perennial grasses, which were fertilized with NPK and are currently under renaturalization. Meanwhile, the highest bulk density (0.28 g cm$^{-3}$) was determined in the former perennial grasses, which were unfertilized and had red clover—timothy mixture. The highest bulk density and the lowest porosity and moisture were identified in the Sapric Drainic Histosol, which is with a removed peat layer

and is under the sward. This was caused by the natural mineralization of the 0–30 cm layer and the resulting shrinkage of the layer. Cultivation of perennial legumes and grasses in Sapric Histosol, which are drained, increased the sustainability of its organic matter and decreased labile carbon content in its profile. This is illustrated by the MHA:MFA ratio of 0.83–0.86 with a recorded in the 0–30 cm top layer. Such a ratio is more favorable for histosol, which is used in agriculture, compared to the same unused Histosol, where this ratio is 0.74.

The E4:E6 ratio data showed that the best quality of humic acids and their greatest maturity is characterful for Sapric Histosol, which is located under former unfertilized perennial grasses and field crop rotation with grasses. Evaluation of the quality of humic acids according to their spectral characteristics (E4:E6) revealed their highest maturity in the former unfertilized perennial grasses and field crop rotation with grasses. Unfertilized perennial grasses and red clover with timothy use promote humification, while fertilized perennial grasses promote mineralization due to the use of mineral NPK fertilizers. Spontaneous renaturalization of drained Sapric Histosol promotes peat mineralization due to the lack of moisture in the peat soil. These data show that to achieve sustainable use of Sapric Histosol and reduction of GHG emissions in peatlands, controlled renaturalization with targeted seeding grasses, especially species of leguminous, must be carried out.

The quality of organic matter at the molecular level changes in Sapric Histosol when the peat layer is removed. Analysis of molecular fractions and polydispersity index in the context of organic matter quality showed that aromatic and phenolic groups, which have hydrophobic properties, are more protected against mineralization. This is established in Sapric Histosol, which is affected by renaturalization, using species: red clover-timothy mixture and crop rotation with perennial grasses.

Hydrophobic (aliphatic and aromatic rings) and hydrophilic bonds were equally expressed in the Sapric Histosol with a non-removed peat layer, while only hydrophilic bonds in the forest Sapric Histosol with a removed peat layer were expressed.

From an agroecological point of view, humic substances are significantly more sustainable and resistant to mineralization in that drained Sapric Histosol where the peat layer has not been removed, and it is used as a perennial grassland.

**Author Contributions:** Conceptualization, K.A.-V. and J.V.; Methodology, K.A.-V., J.V., J.C., I.M. and V.L.; Software, K.A.-V. and J.V.; Validation, K.A.-V. and J.V.; Formal analysis, K.A.-V., J.V., R.P. and V.L.; Investigation, K.A.-V., J.V. and A.S.; Resources, K.A.-V. and J.V.; Data curation, K.A.-V., J.V., J.C. and R.P.; Writing—original draft, K.A.-V., J.V. and J.C.; Writing—review & editing, K.A.-V. and J.V.; Visualization, K.A.-V. and J.V.; Supervision, K.A.-V. and A.S. All authors have read and agreed to the published version of the manuscript.

**Funding:** This research received no external funding.

**Institutional Review Board Statement:** Not applicable.

**Informed Consent Statement:** Not applicable.

**Data Availability Statement:** Not applicable.

**Acknowledgments:** This study was conducted in compliance with the long-term research program "Productivity and sustainability of agricultural and forest soils" implemented by the Lithuanian Research Centre for Agriculture and Forestry.

**Conflicts of Interest:** The authors declare no conflict of interest.

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
