# Peer review of "The Impact of Profile Genesis and Land Use of Histosol on Its Organic Substance Stability and Humic Acid Quality at the Molecular Level"

_sustainability, doi:10.3390/su15075921_

Round 1

Reviewer 1 Report

please find comments on attache file.

Author Response

Thank you for your review. We accept all your comments and we have made corrections in manuscript. Below you could find answers to yours comments:

Reviewer’s comment:

Authors' response:

1.      HA -please define abbreviations before using it.

To reviewer's comment has been considered and corrected.

2.      DOM please define abbreviations before using it.

To reviewer's comment has been considered and corrected.

3.      please define abbreviations before using it.

To reviewer's comment has been considered and corrected.

4.      any evidence can support this?

To reviewer's comment has been considered and corrected.

5.      Keywords please keep keywords limited to 5.

To reviewer's comment has been considered and corrected.

6.      also, the figure is too complex to be graphical abstract. it also contain abbreviations which are not predefined.

We partially agree with the comment. It will be read by specialists in this field, abbreviations are widely known in the international literature, so they should not complicate misunderstanding. We also consider an abstract to be comprehensive, not complicated.

7.      Introduction sentence can not start with and

To reviewer's comment has been considered and corrected.

8.      Introduction the introduction is too long. It feels like a list of already known literature. However, it does not include what is missing in existing literature. Please strengthen missing point in literature.

The article is complex and under consideration many indicators, which is why the literature review is long. To the reviewer's note has been considered and supplemented with novelty and relevance.

9.      Object 0-30 cm?

The average thickness of a layer that is formed due to tillage, in Lithuania.

10.   Object please either make a table or write this information in a text format not list.

To reviewer's comment has been considered and corrected.

11.   Soil bulk density hese are already widely known formulas. please remove them and add methods names and references for each methods

The reviewer's suggestion was considered, and the citation added:

Lal. R., Shukla M.K. Principles of soil physics.  Ohio, U.S.A., 2004, p. 124-147;

Kadžienė G., Feiza V Determination of soil water content by sorption (pF) method. Scientific methodologies for innovative research in land and forest sciences. Kaunas, Lithuania, 2013, p. 311-327)

12.   Methods lease do not write these names in italic and bold or make them different paragraphs

To reviewer's comment has been considered and corrected.

13.   Statistics please check stat. I do not think these methods can measure this statistical analysis

To reviewer's comment has been considered and supplemented „Significant differences among treatment means were assessed by Fisher’s test, where p-values were calculated, and a value of p < 0.05 was considered statistically significant “

14.   Discussion the discussion is short compare to other sections. There are numerous sections in results not blended in discussion well

We would like to disagree on this point. Therefore, for the sake of discussion, we would like to express our position. We agree with the reviewer's observation that, at first glance, the discussion section appears too short. However, this is not really the case. Indeed, all the results of the study are discussed/discussed. In the discussion section, we have placed the essential part of the discussion of the research results, i.e. the discussion of the qualitative parameters of humus materials. These are the results that are directly related to the article title. We put them here to make it a highlight. Meanwhile, the results of the analysis of peatland profiles remained interspersed with discussion points in the results section. We deliberately chose this way of presenting results. The analysis of peat soil profiles is a complex process, so this part is partly a description of the research object and the result and discussion. Separating the discussion of the results of the peatland profile from the results can make the results themselves difficult to understand. In our article, we would like to continue with the chosen way of presenting the results and discussion.

15.   Conclusions please do not write this section as a list.

To reviewer's comment has been considered and corrected.

Reviewer 2 Report

The authors investigated different land use practices on the soil properties including humic acid quality. The results might be helpful to understand the cycle and stability of C in soil under the influence of dramatically different agricultural practices. However, the manuscript has great room to improve, particularly in its writing and language. Some sentences are not complete, and the current manuscript is too lengthy, which frustrates readers to get the results that the authors may want to present. For instance, the Methods and Conclusions sections can be greatly compressed. Where the authors conducted the analyses is not important to readers and should be deleted. In contrast, reliable analysis methods matter to the results.

Some specific suggestions:

It is better to provide a reference to support the statement: “Hydrophobic of peat describes a condition where soil can no longer absorb water and irreversible dries. ”

The last four paragraphs in the Introduction section should be rewritten (or combined) to make it concise and clear.

Could the H2SO4 digest the soil completely to be competent for the total N, P or K analysis? Did the authors compare this method with classic methods, for example, digestion using aqua regia?

How did the author treat the soil before they determined the contents of SOC in the soils? No specific method or equipment was provided.

How did FTIR spectrum determine the hydrophobicity of humic acids? It is known that FTIR spectroscopy could reflect the distribution of organic functional groups in soil. The information on how the authors calculated the hydrophobicity based on the organic functional group composition should be provided.

Figure 7. What sample did the authors show in this figure, dissolved humic acids extracted from the soil samples? The statement “the Sapric Histosol samples ” is too general. Specific information about the component should be provided. In addition, how did the authors differentiate DOM from LMW compounds? To my knowledge, DOM also contains high amounts of LMW compounds.

Author Response

We sincerely thank you for the review. We agree with the comments and have corrected the manuscript.

Below you will find the answers to your comments:

Reviewer's comment:

Authors' response:

1. Methods and Conclusions sections can be greatly compressed.

according to reviewer's comment, the section has been shortened.

2. It is better to provide a reference to support the statement: “Hydrophobic of peat describes a condition where soil can no longer absorb water and irreversible dries. ”

The sentences have changed based on a reviewer's comment.

3. The last four paragraphs in the Introduction section should be rewritten (or combined) to make it concise and clear

Based on a reviewer's comment, the abstract has been revised.

4. Could the H2SO4 digest the soil completely to be competent for the total N, P or K analysis? Did the authors compare this method with classic methods, for example, digestion using aqua regia?

Chemical analysis of soils were carried out using approved methods at the Chemical Research Laboratory of the Institute of Agriculture, Lithuanian Research Center for Agriculture and Forestry.

5. How did the author treat the soil before they determined the contents of SOC in the soils? No specific method or equipment was provided.

Based on reviewer’s comment, the description of preparation of samples were supplemented.

6. How did FTIR spectrum determine the hydrophobicity of humic acids? It is known that FTIR spectroscopy could reflect the distribution of organic functional groups in soil. The information on how the authors calculated the hydrophobicity based on the organic functional group composition should be provided.

We used an integral intensity of O-H stretch region and integral intensity of both C=O stretch bands to characterize hydrophilicity of the soil samples the ratio between the bands of highly hydrophobic C-H groups, and hydrophilic C=O groups.

7. Figure 7. What sample did the authors show in this figure, dissolved humic acids extracted from the soil samples? The statement “the Sapric Histosol samples” is too general. Specific information about the component should be provided.

Based on reviewer’s comment, the title of 7 fig.  has been corrected. “Water extractable DOM molecular fractions of Sapric Histosol with a non–removed peat layer and Sapric Histosol with a removed peat layer by HPLC (%)”

8. In addition, how did the authors differentiate DOM from LMW compounds? To my knowledge, DOM also contains high amounts of LMW compounds.

The chromatograms were recorded and processed by the Agilent ChemStation software and it's library.

Reviewer 3 Report

Article Type: Research article

Title: The impact of profile genesis and land use of Histosol on its organic substance’s stability and humic acids quality in molecular level

 The subject of the manuscript is intriguing, and it falls within the scope of the journal of Sustainability. Following is my evaluation of the paper:

Abstract requires a more articulate style of writing and the aim should be clearer.

Introduction section requires a more articulate style of writing as well as this section should provide a more precise evaluation of the significance of the study. The Authors have to check how references should be used.

 I missed a reference of formula and statistical analysis.

All list of references should be rewritten in accordance with the requirements of the journal

I want to pay attention of 35 and 40 surname writing.

The authors should check how should be writing CO2 and etc.

Decision: Paper can be published with major revisions.

Author Response

Thank you very much for your review. We agree with the comments and revised the manuscript.
Below you will find the answers to your comments:

Reviewer's comment

Authors' response:

1.      Abstract requires a more articulate style of writing and the aim should be clearer.

The abstract is supplemented with an objective.

2.      Introduction section requires a more articulate style of writing as well as this section should provide a more precise evaluation of the significance of the study. The Authors have to check how references should be used.

The importance of the study has been highlighted and the introduction supplemented, as well as the last four paragraphs have been revised.

3.      I missed a reference of formula and statistical analysis.

Based to reviewer's comment reference of formula and statistical analysis were considered and supplemented.

4.      All list of references should be rewritten in accordance with the requirements of the journal

To reviewer's comment has been considered and corrected.

5.      I want to pay attention of 35 and 40 surname writing.

The spelling of the authors' names is as it appears in the journal.

6.      The authors should check how should be writing CO2 and etc.

To reviewer's comment has been considered and corrected.

Round 2

Reviewer 2 Report

The authors have made a good revision to the manuscript according to my comments. I have no more other suggestions.

Reviewer 3 Report

I think that this manuscript  may be suitable for the publication on Sustainability